# Evaluation of a community health worker home visit intervention to improve child development in South Africa: A cluster-randomized controlled trial

**Peter C. Rockers** [1] *, **Jukka M. Leppänen** [2], **Amanda Tarullo** [3], **Lezanie Coetzee** [4], **Günther Fink** [5,6], **Davidson H. Hamer** [1,7], **Aisha K. Yousafzai** [8], **Denise Evans** [4]

1 Department of Global Health, Boston University School of Public Health, Boston, Massachusetts, United States of America, 2 Department of Psychology and Speech-Language Pathology, University of Turku, Turku, Finland, 3 Department of Psychological and Brain Sciences, College of Arts and Sciences, Boston University, Boston, Massachusetts, United States of America, 4 Health Economics and Epidemiology Research Office, Faculty of Health Sciences, University of the Witwatersrand, Johannesburg, South Africa, 5 Swiss Tropical and Public Health Institute, Basel, Switzerland, 6 University of Basel, Basel, Switzerland, 7 Section of Infectious Diseases, Department of Medicine, Boston University Chobanian & Avedisian School of Medicine, Boston, Massachusetts, United States of America, 8 Department of Global Health and Population, Harvard T.H. Chan School of Public Health, Boston, Massachusetts, United States of America

* prockers@bu.edu

## Abstract

### Background

Effective integration of home visit interventions focused on early childhood development into existing service platforms is important for expanding access in low- and middle-income countries (LMICs). We designed and evaluated a home visit intervention integrated into community health worker (CHW) operations in South Africa.

### Methods and findings

We conducted a cluster-randomized controlled trial in Limpopo Province, South Africa. CHWs operating in ward-based outreach teams (WBOTs; clusters) and caregiver–child dyads they served were randomized to the intervention or control group. Group assignment was masked from all data collectors. Dyads were eligible if they resided within a participating CHW catchment area, the caregiver was at least 18 years old, and the child was born after December 15, 2017. Intervention CHWs were trained on a job aid that included content on child health, nutrition, developmental milestones, and encouragement to engage in developmentally appropriate play-based activities, for use during regular monthly home visits with caregivers of children under 2 years of age. Control CHWs provided the local standard of care. Household surveys were administered to the full study sample at baseline and endline. Data were collected on household demographics and assets; caregiver engagement; and child diet, anthropometry, and development scores. In a subsample of children, electroencephalography (EEG) and eye-tracking measures of neural function were assessed at a lab concurrent with endline and at 2 interim time points. Primary outcomes were as follows:

**Funding:** This study was funded by the South African Medical Research Council (www.samrc.ac.za) through a grant to the Health Economics and Epidemiology Research Office at the University of the Witwatersrand (received by DE). The funder had no role in study design, data collection and analysis, decision to publish, or preparation of the manuscript.

**Competing interests:** The authors have declared that no competing interests exist.

**Abbreviations:** aMD, adjusted mean difference; aOR, adjusted odds ratio; CHW, community health worker; CI, confidence interval; COVID-19, Coronavirus Disease 2019; EEG, electroencephalography; HAZ, height-for-age z-score; ICC, intracluster correlation coefficient; LMIC, low- and middle-income country; MD, mean difference; MDAT, Malawi Developmental Assessment Tool; OR, odds ratio; P-CR, pupil-corneal reflection; SD, standard deviation; SRT, saccadic reaction time; WBOT, ward-based outreach team.

height-for-age z-scores (HAZs) and stunting; child development scores measured using the Malawi Developmental Assessment Tool (MDAT); EEG absolute gamma and total power; relative EEG gamma power; and saccadic reaction time (SRT)—an eye-tracking measure of visual processing speed. In the main analysis, unadjusted and adjusted impacts were estimated using intention-to-treat analysis. Adjusted models included a set of demographic covariates measured at baseline. On September 1, 2017, we randomly assigned 51 clusters to intervention (26 clusters, 607 caregiver–child dyads) or control (25 clusters, 488 caregiver–child dyads). At endline (last assessment June 11, 2021), 432 dyads (71%) in 26 clusters remained in the intervention group, and 332 dyads (68%) in 25 clusters remained in the control group. In total, 316 dyads attended the first lab visit, 316 dyads the second lab visit, and 284 dyads the third lab visit. In adjusted models, the intervention had no significant impact on HAZ (adjusted mean difference (aMD) 0.11 [95% confidence interval (CI): −0.07, 0.30]; $p = 0.220$) or stunting (adjusted odds ratio (aOR) 0.63 [0.32, 1.25]; $p = 0.184$), nor did the intervention significantly impact gross motor skills (aMD 0.04 [−0.15, 0.24]; $p = 0.656$), fine motor skills (aMD −0.04 [−0.19, 0.11]; $p = 0.610$), language skills (aMD −0.02 [−0.18, 0.14]; $p = 0.820$), or social–emotional skills (aMD −0.02 [−0.20, 0.16]; $p = 0.816$). In the lab subsample, the intervention had a significant impact on SRT (aMD −7.13 [−12.69, −1.58]; $p = 0.012$), absolute EEG gamma power (aMD −0.14 [−0.24, −0.04]; $p = 0.005$), and total EEG power (aMD −0.15 [−0.23, −0.08]; $p < 0.001$), and no significant impact on relative gamma power (aMD 0.02 [−0.78, 0.83]; $p = 0.959$). While the effect on SRT was observed at the first 2 lab visits, it was no longer present at the third visit, which coincided with the overall endline assessment. At the end of the first year of the intervention period, 43% of CHWs adhered to monthly home visits. Due to the COVID-19 pandemic, we were not able to assess outcomes until 1 year after the end of the intervention period.

## Conclusions

While the home visit intervention did not significantly impact linear growth or skills, we found significant improvement in SRT. This study contributes to a growing literature documenting the positive effects of home visit interventions on child development in LMICs. This study also demonstrates the feasibility of collecting markers of neural function like EEG power and SRT in low-resource settings.

## Trial registration

PACTR 201710002683810; https://pactr.samrc.ac.za/TrialDisplay.aspx?TrialID=2683; South African Clinical Trials Registry, SANCTR 4407

## Author summary

### Why was this study done?

- Prior studies have demonstrated that home visit interventions delivered by community health workers (CHWs) can have a positive impact on child development.

- We integrated a home visit intervention into existing CHW operations in South Africa to generate evidence useful for local decision-making.

- A key innovation of this study was the inclusion of 2 markers of neural function as primary outcomes: electroencephalography (EEG) power and saccadic reaction time (SRT), an eye-tracking measure of visual processing speed.

## What did the researchers do and find?

- We conducted a cluster-randomized controlled trial in rural South Africa. Intervention CHWs were trained on a job aid that included content focused on improving child development, for use during regular home visits with caregivers and young children. Control CHWs provided the local standard of care.

- We enrolled 607 caregiver–child dyads in 26 intervention clusters and 432 dyads in 25 control clusters at the start of the study. At endline, we assessed 432 intervention dyads (71%) and 332 control dyads (68%) to evaluate impact on child development outcomes.

- The home visit intervention decreased child SRT by 7 milliseconds, indicating improved neural function, but did not significantly impact linear growth or skills.

## What do these findings mean?

- Our findings contribute important new evidence to a growing literature documenting the positive effects of home visit interventions on child development.

- Collecting markers of neural function in low-resource settings is feasible.

## Introduction

A strong body of evidence demonstrates that the negative impacts of early adversity on child development in low- and middle-income countries (LMICs) can be mitigated through appropriate early life interventions [1]. Regular home visits by trained staff who support and counsel caregivers on a diversity of topics, including child health, nutrition, and play-based stimulation, have been shown to be particularly effective [2]. However, important questions remain around how to effectively integrate home-based interventions for children into existing service platforms in order to deliver them at scale [3,4].

   Community health workers (CHWs) in LMICs are increasingly being tasked with implementing home visit programs focused on child development. Evidence of the impact of early childhood services delivered by CHWs on child health and development is mixed, and in many settings, CHWs are overburdened and lack supportive supervision [5]. In South Africa, CHW home visits are an important part of the National Integrated Early Childhood Development Policy that the government aims to fully implement over the next decade [6]. CHWs in South Africa are responsible for a broad range of community-based services, including care for HIV and TB patients, and effectively integrating child development services into their set

of responsibilities requires careful planning, including establishing a manageable schedule and ensuring adequate support, to avoid additional overburdening. We conducted a cluster-randomized controlled trial to test the impact of a CHW-delivered home visit intervention on child development in Limpopo Province, South Africa. The intervention was integrated into existing CHW operations in order to generate evidence that would be useful for local decision-making related to potential scale-up. CHWs in the study area operate as part of ward-based outreach teams (WBOTs), and a cluster design with WBOTs as clusters was used to minimize potential spillovers between study arms.

Trials testing home visits and related early childhood interventions in LMICs have predominantly used linear growth and assessments of children's skills as primary outcomes. Recent reviews of the science of child development have identified the need to include "direct assessments of the brain" in trials, in part to improve understanding of the neural mechanisms through which early interventions affect skill formation [2,7]. A key innovation of this study is the inclusion of 2 markers of neural function as primary outcomes: electroencephalography (EEG) power and saccadic reaction time (SRT), an eye-tracking measure of visual processing speed. Basic science research conducted with animal models and human populations over the past several decades has established that the aspects of the brain captured by EEG power and SRT are integral to the performance of essential cognitive functions, including attention, memory, and sensory motor coordination [8,9]. These cognitive functions are the foundation upon which behavioral and academic skills are built [10,11].

We tested a primary hypothesis that a CHW-delivered home visit intervention would improve child development in South Africa.

## Methods

### Study design and setting

The study was a cluster-randomized controlled trial implemented in the catchment areas of 51 CHW WBOTs in Greater Tzaneen and Greater Giyani subdistricts, Mopani District, Limpopo Province, South Africa. This trial was registered in the South African Clinical Trials Registry (SANCTR 4407) and the Pan-African Clinical Trials Registry (PACTR 201710002683810; https://pactr.samrc.ac.za/TrialDisplay.aspx?TrialID=2683) prior to baseline data collection. We describe the study and our findings in this paper accordance with the CONSORT checklist (S1 CONSORT Checklist). The full protocol is provided in S1 Text.

### Participants

All CHWs who were part of WBOTs operating in Greater Tzaneen and Greater Giyani and met the following eligibility criteria were invited to participate in the study: active as a CHW for more than 3 months at the time of enrollment; at least 18 years of age; completed the standard Phase I training for CHWs in South Africa; not using mHealth technologies as part of routine activities; not participating in another research project; and had an eligible caregiver–child dyad in their catchment area. Nearly all CHWs operating in the study area were female. WBOTs are units of 4 to 10 CHWs working together, linked to a primary care facility, and supervised by an Outreach Team Leader, usually a professional nurse [12]. The South Africa National Department of Health has established WBOTs as a foundational component of their primary healthcare system, and CHWs operating within WBOTs are paid by the government. Typically, each CHW is responsible for providing community-based services to around 250 households. As part of standard CHW protocols, all children under the age of 2 should be visited at home once per month. Based on the demographics of the study area, each CHW was

responsible for making visits to around 60 children under the age of 2. Preliminary work indicated that CHW home visits for young children generally lasted 30 minutes.

Caregiver–child dyads were eligible for the study if they resided within a participating CHW catchment area, the caregiver was at least 18 years of age, and the child was born after December 15, 2017. This date was chosen so that all study children were born after the initial training of intervention CHWs and thus exposed to the intervention from birth. Caregivers who were unwilling to provide informed consent or who planned to move from their CHW catchment zone during the period of the study were excluded. Eligible dyads were identified through active monitoring by CHWs, whose responsibilities include supporting pregnant women with accessing antenatal services provided at clinics, and review of birth registries at public hospitals and clinics in the study area. According to the South Africa Demographic and Health Survey 2016 [13], 98% of births in Limpopo occur at health facilities, and the vast majority are at public clinics and hospitals. The research team visited eligible dyads at home where they were recruited and enrolled in the study. CHWs received 10 Rand of mobile phone airtime for each eligible dyad identified. The recruitment process continued until the target sample size was reached. The youngest child enrolled in the study was born on March 15, 2018.

After enrollment was completed, all study dyads residing in Greater Tzaneen subdistrict were invited to participate in assessments of neural function conducted at a lab centrally located in the town of Tzaneen, the capital of the subdistrict. Children with low birthweight ($<2.5$ kg) or a diagnosed developmental disorder were excluded from the lab subsample due to an expectation of insufficient statistical power to appropriately examine atypical development. Eligibility for the lab subsample was restricted to 1 subdistrict due to budget constraints.

The study was approved by the Human Research Ethics Committee at the University of the Witwatersrand (#M160251, #M180229, #M210570) and by the Institutional Review Board at Boston University (#H-37065) prior to the enrolment of participants. All caregivers provided written informed consent prior to the start of the study and at each lab visit. Caregivers received 150 Rand (approximately US$10) at each study visit.

## Randomization and masking

CHWs and the dyads they served were randomized at the level of WBOT (cluster) to either the intervention or control group by a member of the investigator team (PCR). The randomization was performed prior to the start of CHW training and only CHWs in the intervention arm were trained on the study job aid. In 2 instances where more than 1 WBOT operated out of the same health facility, WBOTs were grouped together in the same study cluster to reduce the risk of contamination. Cluster assignment was determined using a covariate constrained randomization procedure that maximized balance on number of health facilities, average monthly volume of under-5 consultations at health facilities over the past year, number of WBOTs, number of CHWs, average CHW age, education, and years in the position. As part of an exploratory aim that is outside the scope of this paper, intervention WBOTs were then randomized a second time using the same covariate constrained randomization procedure to receive one of 2 versions of the CHW job aid, which differed only with respect to the suggested play-based activities. Group assignment was masked from all data collectors, but masking of participating CHWs and dyads was not possible.

## Procedures

**Intervention design.** Intervention CHWs were provided with and trained on a job aid that included content on child health, nutrition, developmental milestones, and

encouragement to engage in developmentally appropriate play-based activities. The full TIDieR Checklist is provided in S1 Table. They were instructed to use the job aid to structure and guide the content of their regular monthly home visits with children under 2 years of age and their caregivers. CHWs did not receive payment for participating in the study. They were provided with the job aid and a carry bag and received payment to cover transportation to attend trainings. The job aid included 24 pages, one for each month from birth to age 2, with appropriate content for the child's month-age. The pages for the first 6 months of the child's life included additional content focused on maternal well-being. For example, the 1-month visit included asking mothers how they were feeling and referring them to appropriate resources if indicated. During training, CHWs were instructed to flip to the page that matched the age of the child they were visiting. The job aid was designed so that the caregiver and child could view an outward facing page that included simple messages in 3 local languages (English, Sepedi, and Xitsonga) and corresponding illustrations of visit content. Fig 1 presents an example of a caregiver-facing page from the CHW job aid. On the CHW-facing side of the job aid was a flowchart with the same content but with more detail that guided the CHW through the visit. The content of the job aid related to developmental milestones was also summarized in a

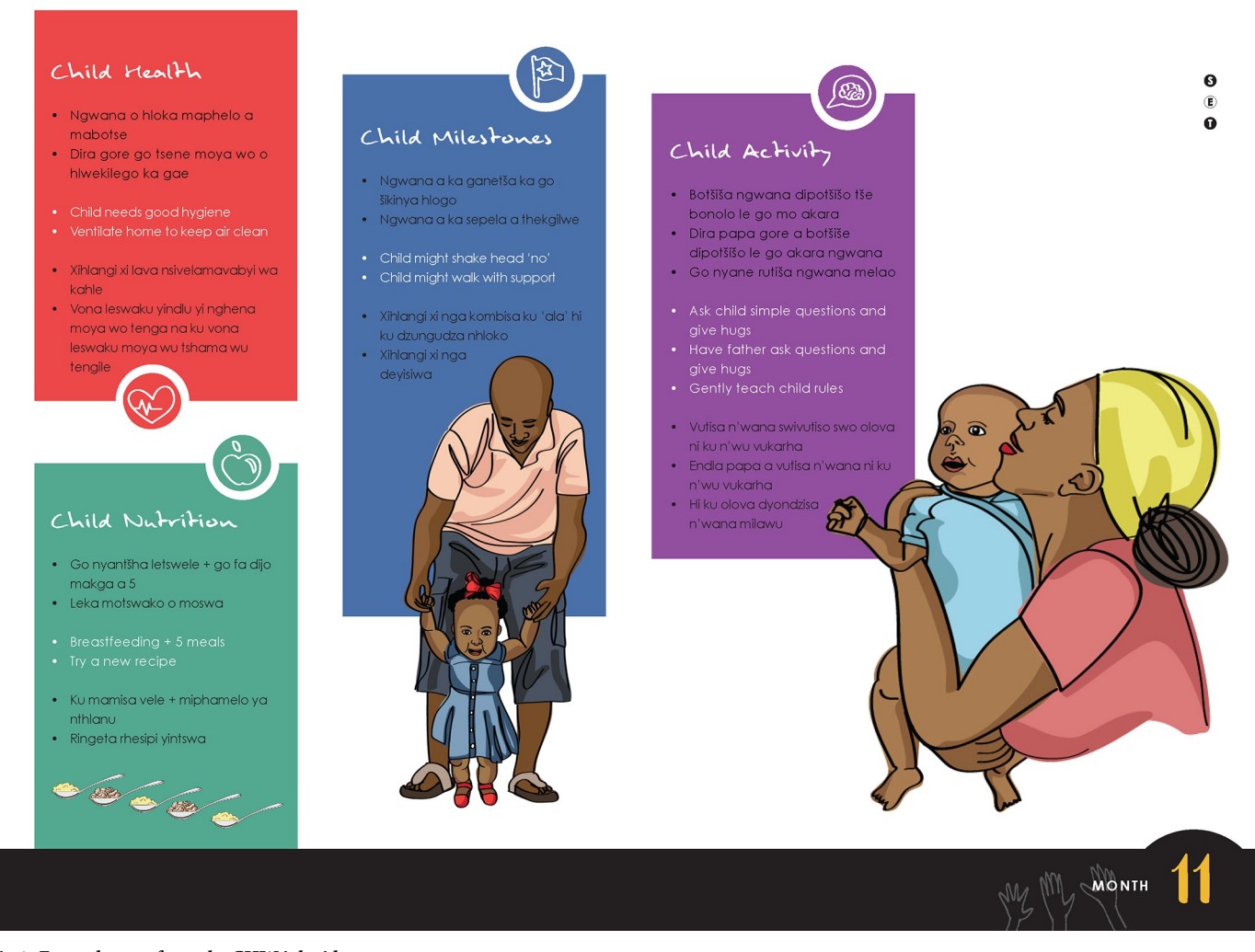

**Fig 1. Example page from the CHW job aid.**

small poster that was provided to each household for caregivers to keep and refer to throughout the course of the intervention period. A second poster containing information on WHO growth standards was also provided to each household.

The content and structure of the job aid were developed over the course of the year that preceded the start of the trial through a process that included several steps. Content development was informed by a robust body of evidence demonstrating that home visits covering a diversity of topics, including child health, nutrition, and play-based stimulation, can generate significant positive effects on child development [2]. The structure of the job aid built on recent research showing that implementation factors, including design aspects of curricula, training manuals, and job aids, are critical in determining the effectiveness of home visit interventions [14]. First, we reviewed the content of similar home visit tools used in previously published intervention studies or by non-governmental organizations supporting or operating home visit programs. Some of the tools we reviewed were available in the public domain, and others were obtained through request. Next, we built a master list of content from all of the reviewed tools, organized according to our preselected domains, i.e., health, nutrition, milestones, and play-based activities, and by age appropriateness. Then, we reviewed the master list with a local advisory committee, which included the head pediatrician of the Limpopo Department of Health and the head supervisor of CHWs operating in Mopani District. As part of this review, we cut the list down to include no more than 3 messages or activities per domain per age-month. At this time, we also aligned messaging with the South African Road to Health Booklet, an important resource that CHWs commonly refer to during regular home visits. Next, we engaged a local graphic artist to design the job aid and add appropriate pictorial representations that corresponded to the content. This resulted in an initial draft of the full job aid, which we then refined through additional consultation with the local advisory committee and through piloting with local CHWs.

Two versions of the job aid were developed that differed only with respect to the suggested play-based activities and the approach to how caregivers interacted with their child during play. The first version emphasized caregiver-directed interactions focused on developing and reinforcing skills within the child's zone of proximal development [15]. For example, on the month-12 page of this version, the suggested activities included "take the child for walk and talk to the child about nature." The second version encouraged caregiver-child interactions focused on strengthening responsiveness [16]. On the month-12 page of this version, the suggested activities included "respond to child's attempts to talk with a smile and a hug." During home visits, CHWs encouraged mothers to try the activities described in the job aid and then provided feedback. Full versions of both job aids are provided in S1 Fig; additional intervention resources are available from the corresponding author upon request. Our aim in developing and using 2 versions of the job aid was to explore the relative importance of strengthening the caregiver–child relationship with a greater focus on responsiveness versus strengthening child skills with a greater focus on developmental stimulation.

Participants in the control arm received the local standard of care throughout the study period. CHWs in the study area are responsible for making monthly home visits for all children under the age of 2 and for providing a broad range of services during those visits, including health screening, immunization monitoring, and nutrition counselling. The Road to Health Booklet served as a guide for control CHW visits as well as a record of the child's growth and vaccination status. It also provided caregivers with important information on child health, nutrition, and common developmental milestones. Intervention CHWs continued to use the Road to Health Booklet to record child growth and vaccination status, and the key health promotion messages in the Road to Health Booklet were integrated into the job aid. The intervention job aid expanded the content of home visits but was designed to be effort neutral by restructuring the time CHWs were already responsible for allotting.

CHWs were trained on the job aid prior to the start of the intervention period and were provided with refresher trainings every 6 months thereafter, amounting to 4 trainings in total over the 2-year intervention period. The first training lasted 4 days, while the subsequent refresher trainings lasted 2 days each. Trainings were conducted by members of the research team with logistical support from a local nongovernmental organization that specialized in training CHWs for the Limpopo Department of Health. Each training also included an overview of key topics related to child development, a subject that was not part of standard CHW training in South Africa at the time of the study. Trainings employed didactic learning methods, which sought to build on and improve the baseline knowledge that CHWs had. Training included live demonstrations of aspects of the curriculum, particularly recommended play-based activities. All CHWs were proficient in English, though many preferred to communicate in their home language (Xitsonga or Sepedi). Trainings were conducted in Sepedi and Xitsonga, which also prepared CHWs to use the job aid in the caregiver's home language. In the 6 months that followed the first training, each CHW was accompanied on a home visit by one of the trainers, who provided additional support and feedback to strengthen CHW knowledge around delivering the intervention.

**Data collection.** Children were exposed to the intervention until they turned 2 years of age, at which time the CHW job aid was no longer used during home visits. The study intervention period ended when the youngest enrolled child turned 2 years of age on March 15, 2020, and endline assessments were originally planned to begin at this time. However, the Coronavirus Disease 2019 (COVID-19) pandemic began to affect South Africa just as fieldwork was set to begin, which required a delay in endline to ensure the safety of participants and study staff. Endline assessments eventually started in September 2020 and were completed in June 2021. Children were around 3 years of age at endline and 1 year removed from the end of the intervention period. The decision to delay endline was made by the study Principal Investigators (PCR and DE), in consultation with all study investigators and the local advisory committee, and in accordance with guidance provided by the study ethics committees. We describe the full impact of these important extenuating circumstances using the CONSERVE-CONSORT checklist (S2 Table).

Household surveys were administered at baseline (January to May 2018) and endline (September 2020 to June 2021). Demographic data and household asset information were collected during the baseline survey. Data on caregiver engagement were collected during baseline and endline surveys. Data on child anthropometry, development, and diet were collected during endline surveys only. EEG and eye-tracking assessments were administered during lab visits during endline (Wave 3: September 2020 to June 2021) and at 2 interim study time points, when children were on average 7 months old (Wave 1: July to November 2018) and when children were on average 17 months old (Wave 2: April to August 2019). Appointments for lab visits were scheduled during phone calls between study staff and caregivers. On the day of the appointment, a study vehicle and driver collected caregivers and children from their home and transported them to and from the lab. Finally, data on implementation of the intervention were collected during a set of phone-based interviews conducted with 1 randomly selected caregiver served by each intervention CHW after the intervention had been running for 1 year. Caregivers were asked to self-report the time since the CHW last visited their home and whether at the last visit the CHW counseled the caregiver on the child development content of the job aid. These interviews were not conducted in the control arm. Our initial plan was to measure implementation factors based on caregiver self-report at endline, but because of the delay in fieldwork due to COVID-19, the recall period was too long to generate quality data.

**Lab assessments.** EEG and eye-tracking assessments were conducted at a lab that was established prior to the start of the study in the town of Tzaneen, which is centrally located within the study area. Each of 2 assessment rooms had one 32-channel portable Geodesic EEG

System 400 (by EGI/Philips Neuro, Eugene, OR) and one Tobii Pro x3-120 120 Hz eye tracker (by Tobii Technology, Stockholm, Sweden). Members of the investigator team with appropriate expertise in EEG (AT) and eye tracking (JL) set up the assessment rooms prior to the start of the study and extensively trained a team of permanent staff on the use of both technologies prior to the first assessment wave and again before each subsequent assessment wave. Assessors did not have previous medical training nor prior experience with the technologies. The average time to complete both assessments, including preparation activities and data collection, was around 45 minutes per child.

When conducting EEG assessments, caregivers were seated on a chair and children were seated on their caregiver's lap. The assessor then entertained the child and provided them with toys while the EEG sensor net was placed on the child's head. The assessor made sure each sensor had a good connection with the scalp. Once impedances below 50 ohms were established for all sensors, the EEG recorded from all channels with reference to the vertex while the child sat on the caregiver's lap with the lights dimmed. EEG data were recorded for 6 minutes. Data were uploaded daily from lab computers to a secure file sharing platform. A member of the investigator team (AT) regularly checked data quality and provided feedback to the lab team.

Pupil-corneal reflection (P-CR) eye-tracking technologies have been used extensively in prior studies with infants and young children [17,18], including in LMIC contexts [19]. P-CR eye trackers are video-based methods that record information about gaze direction based on the coordinates of the pupil and the corneal reflection of an infrared light source in the camera image [20]. Previous studies have confirmed that the expected temporal and empirically verified spatial accuracy of eye-tracking with children is sufficient to reliably quantify SRT [21].

The eye-tracking assessment began just after the EEG assessment was complete. The caregiver remained seated and holding the child on their lap. The caregiver was positioned so that the child's eyes were facing forward and at approximately 60 cm viewing distance from the eye tracker and computer monitor. The caregiver was instructed to turn their head and eyes to the side (approximately 90° from the screen) and to avoid looking at the screen during the assessment. During the assessment, the child was presented with short, alternating blocks of visual stimuli on the monitor that were designed to calibrate the eye tracking system, measure SRTs and fixations to social scenes. A detailed description of the eye tracking assessment is provided elsewhere [21]. Saccade targets were colorful cartoon animations of common objects (e.g., fish, bird, pig, rabbit, or a human face), subtending a 5.7° × 5.7° visual angle. Discernible colorful animations and a novel target for each presentation instead of more uniform visual stimuli were used based on children's known proclivity for novelty and on evidence showing superior response rates to colorful, pictorial stimuli in young children [22]. After an initial stimulus in the center of the screen (not analyzed), the child saw a total of 5 animations in 1 block, and a total of 8 blocks to cumulate enough saccades to reliably estimate mean SRT.

Eye-tracking assessments lasted approximately 10 minutes. Data were uploaded daily from lab computers to a secure file sharing platform. A member of the investigator team (JL) regularly checked data quality and provided feedback to the lab team. Additional details on lab assessment procedures are provided in S2 Text.

At the second and third wave of lab visits children were assessed for autonomic measures of heartbeat and breathing, and, at the third wave, blood samples were collected and tested for biomarkers of systemic inflammation. These data will be analyzed in future research.

## Outcomes

Primary outcomes were as follows: height-for-age z-scores (HAZs) and stunting (HAZ $< -2$); child development scores in the domains of gross motor, fine motor, language, and social

development, measured using the Malawi Developmental Assessment Tool (MDAT); absolute EEG gamma and total power; relative EEG gamma power; and SRT. Standing height was measured at endline by trained enumerators and converted to z-scores using the WHO's Multicentre Growth Standards [23]. The original study protocol indicated that the Bayley Scales of Infant and Toddler Development would be used to assess child development, but the decision was made during the study period to use the MDAT instead. The MDAT is a skill-based measure of child development that was created and validated in a rural southern Africa setting, making it culturally relevant to the study context [24]. The interobserver reliability of the tool is high, with 99% of items having a kappa above 0.75, based on assessments of the same child conducted independently on the same day by 2 observers [24]. The MDAT has been used in recent intervention trails conducted in several countries in sub-Saharan Africa [25–29].

Two nurses with expertise and experience in administering the MDAT traveled from Malawi to Tzaneen where they supported translation of the instrument and conducted an extensive training of a team of study assessors. The MDAT instrument was translated and back translated into Sepedi and Xitsonga prior to administration. For children who attended the lab at endline, MDAT assessments were conducted in the assessment room after EEG and eye-tracking assessments were completed. For children who did not attend the lab, MDAT assessments were conducted at the home at the end of the endline visit. Raw scores of successfully completed items were summed within each domain and then converted to z-scores using the MDAT scoring application [30].

EEG data were first filtered using Net Station software with a 50-Hz notch filter to remove electrical noise and 70-Hz low pass filter. The data were then preprocessed in MATLAB using the Harvard Automated Processing Pipeline for EEG, which was designed to address high levels of artifact or very short recording lengths and is ideal for data collected in settings like rural South Africa [31]. Data from the first assessment wave were segmented into 2-second epochs, while data from the second and third waves were segmented into 1-second epochs. A wavelet-enhanced independent components analysis approach was used to correct for artifact while retaining the entire length of the data file. This approach removes multiple classes of artifact, including blinks, eye movements, drift, or muscle artifact. Bad channels were replaced using spherical spline interpolation, and data were re-referenced to the average reference. Following independent components analysis, observations with more than 15% bad channels, more than 5 interpolated channels, fewer than 40 one-second epochs of usable data out of a possible 360 epochs, or median artifact probability greater than 33% for retained independent components were excluded from the analysis for quality reasons.

EEG power was then computed in MATLAB using the Batch Electroencephalography Automated Processing Platform [32], which is designed to streamline batch EEG processing and increase the reproducibility of results. EEG power is partitioned into frequency bands ranging from slow wave activity (delta) to high frequency oscillations (gamma). The relative concentration of power in low frequencies tends to decrease with age, and high frequency oscillations increase, a trend that derives from normative developmental changes such as neuronal growth and myelination [33]. All EEG power measures were resting-state measures. Resting EEG gamma power provides an index of neural maturity, an indication of whether neurodevelopmental processes are "on track" for age. Gamma power oscillations index synchronization of neuronal firing and are linked to cognition and language in early childhood [34,35]. Channel measures of absolute gamma power per Hz (30 to 48 Hz) were averaged across the entire brain for each assessment at each wave and then log-transformed. In the analysis plan included in the published protocol, only absolute gamma power was identified as a primary outcome using EEG data. Based on the results of that analysis (described below), additional ad hoc analyses of absolute total EEG power and relative gamma power were conducted.

A measures of absolute total EEG power was generated for each assessment at each wave by summing channel measures of power across all frequency bands, averaging across the entire brain, and then log-transforming the result. A measure of relative gamma power was generated for each assessment at each wave by dividing average gamma power across the entire brain by average total power across the entire brain.

SRT is defined as the time interval from the onset of the saccade target, i.e., when a new animation appeared on the monitor, to the first entry of the child's gaze in the area of the saccade target (±0.9˚ margin). SRT estimates were extracted from raw eye-tracking data by an automated script [36]. Briefly, the script applies a median-filter to remove abrupt spikes, merges the xy-coordinates for the 2 eyes by averaging or by using the data of 1 valid eye, and verifies that the recorded saccade reflects a valid gaze shift from the area of the previous saccade target to the area of the new target within the expected time window and is not contaminated by missing or noisy data. Saccades passing the preset validity criteria (10 or more valid saccades; [21]) were used to calculate mean SRT.

Exploratory outcomes included caregiver engagement and child diet. These outcomes were not prespecified in the study protocol and are meant to be hypothesis generating with respect to potential behaviour change pathways through which the intervention could influence child development. Data on caregiver engagement were collected using the 6-item Multiple Indicator Cluster Surveys module [37]. A standardized diet diversity score was constructed using data based on the number of food groups children had consumed the previous day [38]. A household wealth index was created based on principle component analysis of household asset information [39].

The target sample size of 1,092 dyads enrolled across 51 clusters was powered to detect a 0.25 standard deviation (SD) difference in HAZ and MDAT domain z-scores with 80% power assuming 20% of the sample would not be assessed at endline and an intraclass correlation coefficient (ICC) for geographic clusters of 0.05. The target for the lab subsample was 900 assessments across the 3 waves, which provided 80% power to detect a 0.25 SD difference in absolute gamma power and SRT, assuming 85% of assessments would meet the preset validity criteria for inclusion in the analysis, a negligible ICC for geographic clustering, and an ICC for repeated measures within dyads of 0.20.

## Statistical analysis

First, we compared baseline characteristics of participants in the intervention and control groups and characterized attrition by treatment arm. We also examined characteristics by arm among the lab subsample. We then estimated the impact of the intervention on the primary outcomes of interest using an intention-to-treat approach. For HAZ and MDAT domain z-scores, we fit a set of linear mixed models with cluster random effects using Stata's *mixed* package to estimate unadjusted and adjusted treatment effects. For our analysis of SRT and EEG gamma power, we estimated average treatment effects across all 3 waves of assessment by fitting a set of unadjusted and adjusted linear mixed models with cluster and individual random effects, again using Stata's *mixed* package. Unadjusted models included the variables used in the covariate constrained randomization procedure. Adjusted models included an additional set of demographic variables measured at baseline: child age, sex, and number of siblings; household wealth and receipt of Child Support Grant; caregiver age and education; and assessor fixed effects. All models with EEG power data adjusted for data quality variables, including unit/room fixed effects and number of channels with usable data. Similarly, all models with SRT data adjusted for data quality variables, including unit/room fixed effects and number of trials with usable data. For all analyses, data from participants not assessed at endline were assumed to be missing completely at random and a complete-case approach was used.

As a secondary analysis, we fit an interaction model to test if treatment effects on SRT differed across the 3 waves of assessment. We also generated a figure based on marginal predictions from this interaction model. As an ad hoc analysis not specified in our original protocol, we estimated the impact of the intervention on absolute total EEG power and on relative gamma power. We conducted this analysis because the results of the absolute gamma power analysis were counter to our original hypothesis and warranted additional scrutiny. Finally, we conducted a per-protocol analysis, restricting the treatment group to dyads served by adherent CHWs, where adherence was defined as making monthly home visits based on caregiver self-report during phone-based surveys. The full control group was used as the comparison group under the assumption that control CHWs did not have access to the intervention materials or training.

We did not publish a written prospective analysis plan. The specifications of the main models, including the use of linear mixed models with random effects and the selection of control variables to include in adjusted models, were planned prior to endline data collection. The measure of CHW adherence used in the per-protocol analysis was also determined prior to endline data collection. The decision to include absolute total EEG power and relative EEG gamma power as additional outcomes was made after analyzing the impact of the intervention on absolute EEG gamma power in order to better understand those results. A member of the investigator team (GF) conducted the analysis blinded to participant treatment group.

### Dryad DOI

Data were deposited in the Dryad repository: https://doi.org/10.5061/dryad.qnk98sfm2 [40].

## Results

Fifty-one clusters were randomly assigned to intervention (26 clusters, 204 CHWs, 607 caregiver–child dyads) or control (25 clusters, 152 CHWs, 488 caregiver–child dyads). At the end of the study (last assessment June 11, 2021), 432 dyads (71%; 175 dyads not assessed at endline) remained in the intervention group, and 332 dyads (68%; 156 dyads not assessed at endline) remained in the control group (Fig 2). A subsample of 266 dyads in the intervention group and 224 dyads in the control group (490 dyads in total), all residing in the 26 clusters in Greater Tzaneen subdistrict, were recruited for the lab portion of the study. In total, 316 dyads (182 intervention; 134 control) attended the first lab visit, 316 dyads (184 intervention; 132 control) attended the second lab visit, and 284 dyads (164 intervention; 120 control) attended

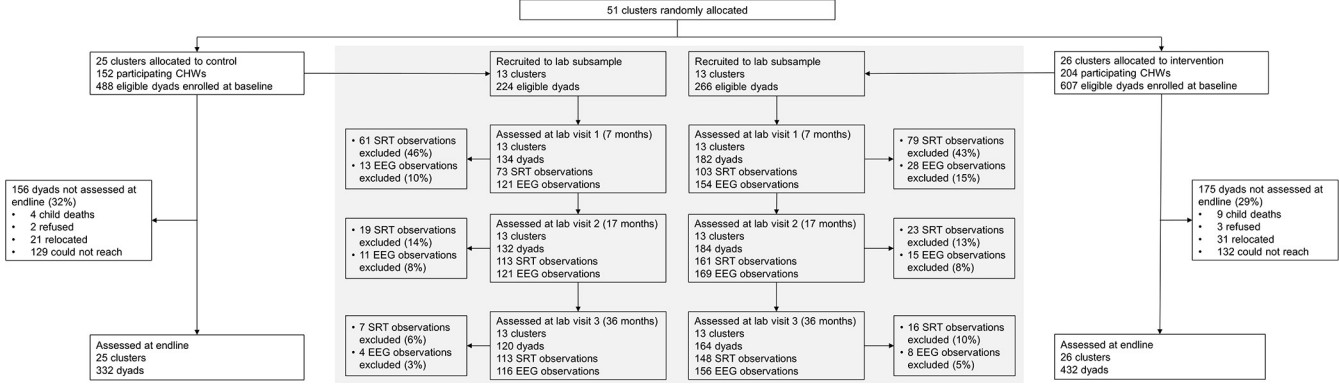

**Fig 2. Trial profile.** CHW, community health worker; EEG, electroencephalography; SRT, saccadic reaction time.

**Table 1. Characteristics of study participants.**

| | Control | | | | Intervention | | | |
|---|---|---|---|---|---|---|---|---|
| | Total Enrolled (n = 488) | Assessed at Endline (n = 332) | Not Assessed at Endline (n = 156) | Assessed at Lab (n = 163) | Total Enrolled (n = 607) | Assessed at Endline (n = 432) | Not Assessed at Endline (n = 175) | Assessed at Lab (n = 223) |
| Child older siblings | 1.47 (1.34) | 1.58 (1.36) | 1.28 (1.30) | 1.44 (1.26) | 1.32 (1.22) | 1.34 (1.19) | 1.24 (1.24) | 1.24 (1.13) |
| Caregiver age at birth (years) | 30.94 (9.00) | 31.56 (8.86) | 32.44 (12.95) | 31. 35 (9.84) | 32.42 (11.33) | 32.41 (10.57) | 29.65 (9.20) | 32.08 (10.66) |
| Caregiver education (years) | 10.04 (2.63) | 10.13 (2.53) | 10.09 (3.01) | 10.22 (2.43) | 10.08 (2.74) | 10.08 (2.61) | 9.86 (2.80) | 10.36 (2.47) |
| Household wealth (z-score) | −0.03 (1.00) | −0.05 (1.00) | 0.07 (1.07) | 0.20 (0.99) | 0.02 (1.00) | 0.01 (0.98) | 0.01 (0.98) | 0.18 (1.02) |
| Household receives Child Support Grant | 0.46 (0.50) | 0.47 (0.50) | 0.43 (0.50) | 0.34 (0.47) | 0.45 (0.50) | 0.46 (0.50) | 0.44 (0.50) | 0.32 (0.47) |

Notes: All values are mean (SD).

the third lab visit. Across the 3 waves of lab assessments, 91% of EEG observations and 78% of SRT observations were determined to meet the preset validity criteria for inclusion in the analysis. There were no clear differences in baseline demographics comparing observations included to those excluded (S3 Table). Overall, 376 eligible CHWs were recruited for the study, of whom 356 (95%) participated. All participating CHWs in the intervention arm attended the first training, while 170 (83%) attended the second training, 183 (90%) attended the third training, and 186 (91%) attended the fourth training.

Caregivers were on average 32 years old and had completed around 10 years of schooling at the start of the study (Table 1). On average, children were 58 days old at the time of enrollment. Just under half of households were recipients of the government Child Support Grant. The study arms were well balanced on key demographics at baseline. In the control group, there was a significant positive association between number of older siblings and probability of being assessed at endline. There were no other significant differences between those assessed and those not assessed at endline. Furthermore, the subsample of participants assessed at the lab differed slightly from the total enrolled sample. In particular, household wealth was higher among those assessed at the lab. However, this was driven by the fact that only households residing in Greater Tzaneen were recruited to attend the lab. The demographics of participants assessed at the lab were very similar to the demographics of the total enrolled sample in Greater Tzaneen.

Controlling for a set of baseline characteristics (Table 2), the intervention had no significant impact on HAZ (adjusted mean difference (aMD) 0.11 [95% confidence interval (CI): −0.07, 0.30]; $p = 0.220$) or stunting (adjusted odds ratio (aOR) 0.63 [0.32, 1.25]; $p = 0.184$), nor did the intervention significantly impact gross motor skills (aMD 0.04 [−0.15, 0.24]; $p = 0.656$), fine motor skills (aMD −0.04 [−0.19, 0.11]; $p = 0.610$), language skills (aMD −0.02 [−0.18, 0.14]; $p = 0.820$), or social–emotional skills (aMD −0.02 [−0.20, 0.16]; $p = 0.816$) assessed using the MDAT. In the lab subsample, the intervention had a significant impact on SRT (aMD −7.13 [−12.69, −1.58]; $p = 0.012$).

The effect on SRT appears to be concentrated in the first 2 waves of assessment and gone at the third wave, which coincided with the full sample endline assessment (Fig 3), though the interaction between the treatment effect and wave of assessment was not statistically significant ($p = 0.193$). The intervention had a significant effect on absolute EEG gamma power (aMD −0.14 [−0.24, −0.04]; $p = 0.005$), and total EEG power (aMD −0.15 [−0.23, −0.08];

**Table 2. Impact on primary outcomes.**

|  | Cluster | Unadjusted[a] |  | Adjusted[b] |  |
|---|---|---|---|---|---|
|  | ICC | MD (95% CI) | *P* value | MD (95% CI) | *P* value |
| **Full sample** |  |  |  |  |  |
| HAZs | 0.030 | 0.18 (0.00, 0.35) | 0.054 | 0.11 (−0.07, 0.30) | 0.220 |
| MDAT z-scores[c] |  |  |  |  |  |
| Gross motor | 0.048 | 0.06 (−0.13, 0.26) | 0.520 | 0.04 (−0.15, 0.24) | 0.656 |
| Fine motor | 0.008 | −0.01 (−0.15, 0.13) | 0.868 | −0.04 (−0.19, 0.11) | 0.610 |
| Language | 0.013 | 0.00 (−0.16, 0.15) | 0.979 | −0.02 (−0.18, 0.14) | 0.820 |
| Social–emotional | 0.004 | −0.03 (−0.21, 0.16) | 0.786 | −0.02 (−0.20, 0.16) | 0.816 |
| **Lab subsample[d]** |  |  |  |  |  |
| Absolute gamma power (ln[$\mu V^2$/Hz]) | <0.001 | −0.13 (−0.22, −0.04) | 0.006 | −0.14 (−0.24, −0.04) | 0.005 |
| Absolute total power (ln[$\mu V^2$/Hz]) | <0.001 | −0.13 (−0.20, −0.06) | <0.001 | −0.15 (−0.23, −0.08) | <0.001 |
| Relative gamma power (percentage) | 0.012 | −0.17 (−0.99, 0.66) | 0.693 | 0.02 (−0.78, 0.83) | 0.959 |
| SRT (milliseconds) | <0.001 | −5.08 (−10.40, 0.24) | 0.061 | −7.13 (−12.69, −1.58) | 0.012 |
|  | Cluster | Unadjusted[a] |  | Adjusted[b] |  |
|  | ICC | OR (95% CI) | *P* value | OR (95% CI) | *P* value |
| **Full Sample** |  |  |  |  |  |
| Stunted | 0.016 | 0.60 (0.31, 1.15) | 0.121 | 0.63 (0.32, 1.25) | 0.184 |

CI, confidence interval; HAZ, height-for-age z-score; ICC, intracluster correlation coefficient; MD, mean difference; MDAT, Malawi Developmental Assessment Tool; OR, odds ratio.

[a]Includes covariates included in the randomization procedure.

[b]Includes covariates included in the randomization procedure and controls: child age, sex and number of siblings; household wealth and receipt of Child Support Grant; and caregiver age and education.

[c]Includes additional covariates indicating location of MDAT assessment (home or lab) and assessor fixed effects.

[d]Estimates based on analysis of panel data from 3 rounds of lab assessment using linear mixed models with cluster and individual random effects and month of assessment fixed effects.

$p < 0.001$). The intervention had no significant impact on relative gamma power (aMD 0.02 [−0.78, 0.83]; $p = 0.959$). In the lab subsample, impacts on MDAT outcomes were similar to those found in the full study sample at endline (S4 Table).

In the full study sample (Table 3), the intervention did not significantly impact diet diversity (aMD 0.19 [−0.04, 0.41]; $p = 0.102$) or caregiver engagement (aMD 0.18 [−0.13, 0.48]; $p = 0.249$). In the lab subsample, there was a significant positive effect on caregiver engagement (aMD 0.33 [0.09, 0.57]; $p = 0.007$).

Based on phone-based survey data collected after the first year of the intervention period, 43% of intervention CHWs were determined to be adherent with monthly home visits, i.e., had made a home visit in the prior month, while 64% had made a home visit in the prior 3 months. Among intervention CHWs who had made a home visit in the prior month, 93% counselled caregivers on the child development content of the job aid. In the per-protocol analysis (Table 4), the intervention had a significant positive impact on HAZ (aMD 0.21 [0.01, 0.41]; $p = 0.039$) and gross motor skills (aMD 0.20 [0.00, 0.40]; $p = 0.045$).

## Discussion

We examined the impact of a CHW home visit intervention integrated into the existing health system in Limpopo Province, South Africa. Our overarching aim was to generate evidence that could inform local policy discussions related to potential scale-up in the country. While the intervention did not have a significant effect on children's linear growth or skills, we found

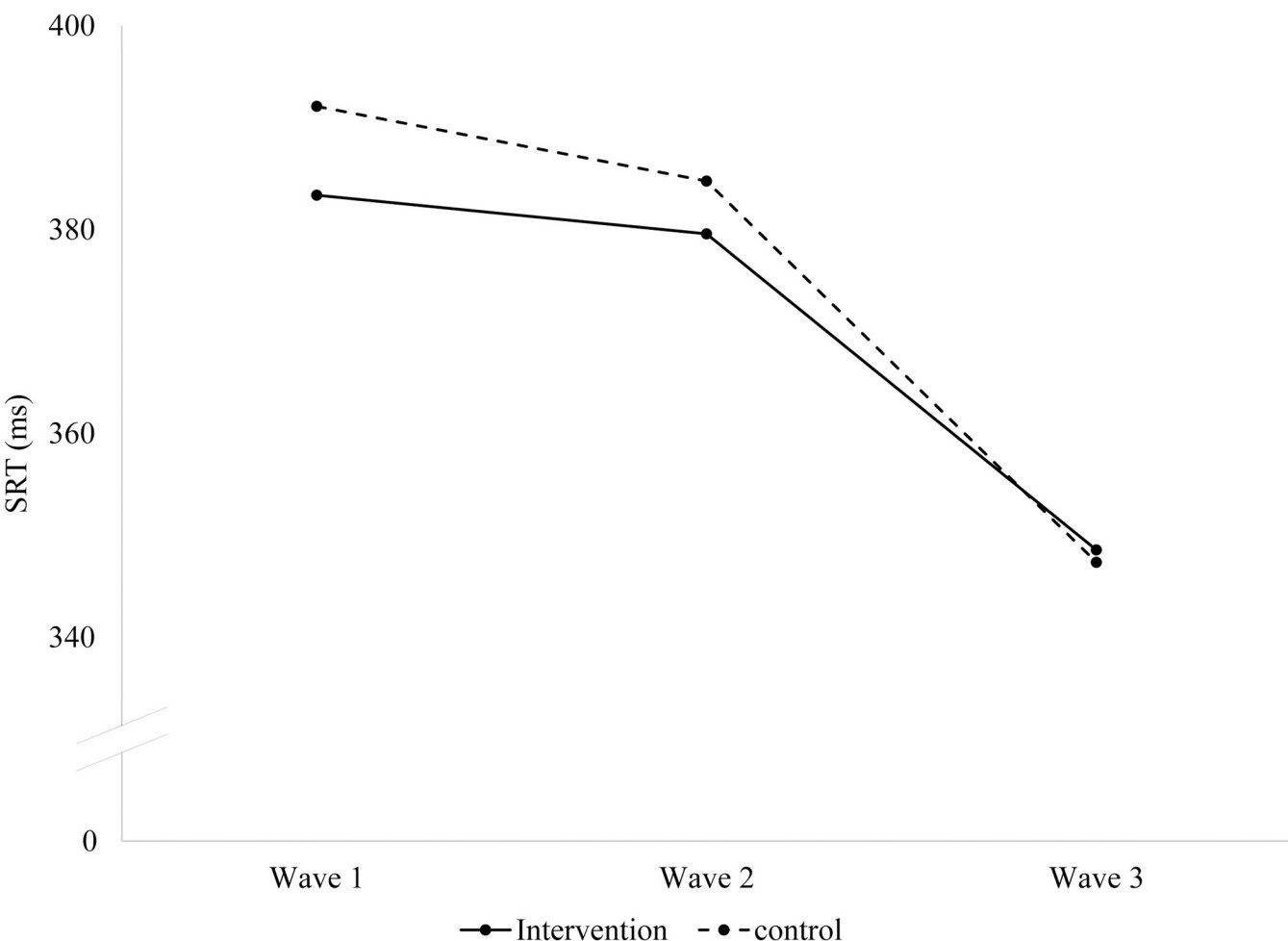

**Fig 3. Impact on SRT by wave. SRT, saccadic reaction time. Notes: Marginal linear predictions based on estimates of treatment–wave interactions from adjusted linear mixed models.** For F-test of hypothesis that treatment effects are equal at all waves: $p = 0.193$. Children were on average 7 months old at Wave 1, 17 months old at Wave 2, and 36 months old at Wave 3. The intervention period ended between Waves 2 and 3, when children were 24 months of age.

**Table 3. Impact on exploratory outcomes.**

| | Cluster | Unadjusted[a] | | Adjusted[b] | |
|---|---|---|---|---|---|
| | ICC | MD (95% CI) | *P* value | MD (95% CI) | *P* value |
| **Full sample** | | | | | |
| Diet diversity | 0.028 | 0.22 (0.01, 0.44) | 0.040 | 0.19 (−0.04, 0.41) | 0.102 |
| Caregiver engagement | 0.027 | 0.22 (−0.09, 0.54) | 0.166 | 0.18 (−0.13, 0.48) | 0.249 |
| **Lab subsample[c]** | | | | | |
| Diet diversity | 0.003 | 0.25 (0.02, 0.47) | 0.032 | 0.22 (−0.01, 0.46) | 0.066 |
| Caregiver engagement | 0.013 | 0.27 (0.01, 0.52) | 0.040 | 0.33 (0.09, 0.57) | 0.007 |

CI, confidence interval; ICC, intracluster correlation coefficient; MD, mean difference.

[a]Includes covariates included in the randomization procedure.

[b]Includes covariates included in the randomization procedure and controls: child age, sex and number of siblings; household wealth and receipt of Child Support Grant; caregiver age and education; and assessor fixed effects.

[c]Estimates based on analysis of panel data from 3 rounds of lab assessment using linear mixed models with cluster and individual random effects and month of assessment fixed effects.

**Table 4. Per-protocol analysis of impact on primary outcomes.**

|  | MD (95% CI) | P value |
|---|---|---|
| **Full sample** |  |  |
| HAZs | 0.21 (0.01, 0.41) | 0.039 |
| MDAT z-scores |  |  |
| Gross motor | 0.20 (0.00, 0.40) | 0.045 |
| Fine motor | 0.15 (−0.02, 0.32) | 0.087 |
| Language | 0.01 (−0.20, 0.21) | 0.946 |
| Social–emotional | 0.12 (−0.12, 0.35) | 0.324 |
| **Lab subsample** |  |  |
| Absolute gamma power (ln[$\mu V^2$/Hz]) | -0.14 (−0.26, −0.02) | 0.025 |
| Absolute total power (ln[$\mu V^2$/Hz]) | −0.16 (−0.28, −0.03) | 0.012 |
| Relative gamma power (percentage) | −0.19 (−1.05, 0.67) | 0.664 |
| SRT (milliseconds) | −7.46 (−18.50, 3.57) | 0.185 |
|  | **OR (95% CI)** | **P value** |
| **Full Sample** |  |  |
| Stunted | 0.40 (0.14, 1.11) | 0.079 |

aMD, adjusted mean difference

CI, confidence interval

HAZ, height-for-age z-score

MDAT, Malawi Developmental Assessment Tool

OR, odds ratio; SRT, saccadic reaction time.

Notes: Adherence defined as CHW having visited home in last month based on data collected during the phone-based surveys. All models include the following covariates: variables included in the randomization procedure; child age, sex and number of siblings; household wealth and receipt of Child Support Grant; and caregiver age and education. MDAT models include additional covariates indicating location of MDAT assessment (home or lab) and assessor fixed effects. Lab subsample estimates are based on analysis of panel data from 3 rounds of lab assessment using linear mixed models with cluster and individual random effects and month of assessment fixed effects.

significant improvement in SRT. Furthermore, we were able to collect high-quality data on neural function using EEG and eye-tracking technologies, confirming that the use of these measures is feasible in low-resource settings.

This study is one of the first in an LMIC to examine the impact of an early life intervention on EEG power and SRT. These technologies generate robust markers of neural function and are portable and well suited for use in low-resource field settings. One recent trial conducted in the United States found that providing households with unconditional cash transfers had a positive impact on infant EEG high frequency power [41]. Another recent trial in Bangladesh found evidence of a positive short-term effect of iron supplementation on EEG alpha power over sensorimotor cortical regions [42]. No significant effects were found for other EEG frequency bands, and the effect on alpha power was not found at a later follow-up assessment. A small pilot study recently conducted in Sierra Leone demonstrated the feasibility of assessing SRT in a low-resource setting and found a relationship between the measure and supplementary feeding in moderately malnourished children [21]. In addition, 2 recent trials in Malawi that tested nutrition interventions found no significant effects on response time for the Infant Orienting with Attention task [43,44], an eye-tracking measure that is similar to our measure of SRT.

Basic science research conducted over the past several decades has identified key pathways through which early adversity affects the brain, establishing a strong scientific basis for

hypothesizing that early interventions that address adversity can positively impact neural function. Nutrient deficiencies can inhibit myelination and lead to defective dendrite growth [45,46]. Inflammation, which can result from infection and malnutrition, has also been linked to hypomyelination as well as decreased cerebral white matter and premature synapse maturation [47–49]. Finally, psychosocial deprivation can compromise neural circuitry [50]. The aspects of the brain captured by EEG gamma power and SRT are integral to the development of important behavioral skills and executive functions [51,52]. Due to their causal proximity, measures of neural function may be more sensitive to intervention effects than behavioral skills like those measured by the MDAT.

Home visits significantly reduced SRT. Lower SRT is indicative of advanced underlying neural maturation and white matter integrity [53,54]. In an analysis we present elsewhere, higher household wealth was strongly associated with lower SRT in the study sample [55]. Previous studies have shown that lower SRT measured during infancy is predictive of stronger cognitive and executive function skills expressed later in childhood [56,57]. Our results indicate that while the effect on SRT was observed at the first 2 lab visits, it was no longer present at the third visit, which coincided with the endline assessment, where we found no effect on HAZ or MDAT scores. One potential explanation for this set of findings is that initial effects that manifested while the intervention was ongoing dissipated during the year after the intervention ended. While our original intention was to conduct the endline assessment at the end of the intervention period in early 2020, the COVID-19 pandemic forced us to delay for 1 year. Several previous studies of early childhood interventions have shown initial benefits that dissipated or disappeared completely as children grew older [58]. Extending the intervention period to age 3 and perhaps even beyond might have generated larger and longer-lasting effects [59]. The difficulties that families and children likely experienced during the first year of the COVID-19 pandemic [60] may have further diminished potential benefits accrued during the intervention period.

Resting EEG gamma power reflects synchronization of neuronal firing, which facilitates the development of efficient neural networks [34,61]. During early childhood, gamma power has been shown to increase with age up to age 4 or 5 years, and higher gamma is often used as a marker of neural maturity [61,62]. The negative impact of the intervention on absolute EEG gamma power is counter to our original hypothesis. The similar effect sizes on absolute gamma and total EEG power—and the corresponding null effect on relative gamma power—suggest an anatomical explanation, possibly related to skull conductivity, rather than evidence of a negative effect on neural function [63]. Anatomical differences may have resulted from dietary changes in the intervention group, though these differences are not necessarily indicative of improved neural function. From a methodological perspective, our findings suggest that using absolute EEG power as an outcome to evaluate interventions that target malnutrition in LMICs may be problematic because the measure may conflate effects on skull conductivity and neural function. Relative power within frequency bands may provide a more reliable measure of underlying neural function. We found no effect on relative gamma power.

We integrated the intervention job aid and training into the existing CHW home visit infrastructure. We did not intervene in the local system of CHW supervision. As a result, important aspects of implementation were not under our control. Most notably, we were not able to ensure that CHWs made monthly visits to study homes. We intended to capture evidence of implementation fidelity at endline based on caregiver reporting, but because of the pause in the study due to COVID-19, the intervention had been completed for 1 year when we eventually conducted endline, making the recall period too long to generate quality data. Based on the phone-based surveillance interviews we conducted during the intervention period, we found that just under half of CHWs had visited the home in the previous month. We found

evidence of significant positive impact on child growth and gross motor skills among households served by adherent CHWs. Poor adherence on the part of CHWs likely diminished the effectiveness of the intervention overall. Recent qualitative work conducted in South Africa suggests that the supervision structure for CHWs is weak [12,64]. While the majority of CHWs in South Africa are well educated—81% of CHWs participating in the study had completed high school—supervision remains important, and supervisors are often responsible for managing the primary care clinic and have limited resources and time to spend with CHWs.

This study had important limitations. First, as discussed above, as a result of the delay in fieldwork due to the COVID-19 pandemic, we were not able to assess children at the end of the intervention period, and we were not able to thoroughly document key aspects of implementation. Second, the EEG-based outcome measure we specified in our protocol, absolute gamma power, may not be the most appropriate EEG-based marker of neural function for trials of interventions that may influence skull conductivity through nutrition channels. The use of EEG data in intervention trials in LMICs holds promise, but more work is needed to identify the most appropriate measures that can be constructed from these complex data for use in these contexts. Finally, likely due in part to the 1-year delay in endline, attrition was higher than was assumed in the initial power calculation (30% versus 20%), leading to a reduction in statistical power. Based on Table 1, there does not appear to be a systematic relationship between attrition and treatment group or participant demographics, supporting the use of the complete-case approach to dealing with missing data in the analysis.

This paper contributes to a growing body of evidence documenting the positive effects of home visit interventions on child development in LMICs [2]. Integrating such interventions into the existing health system in South Africa appears feasible. In addition, this study demonstrates the feasibility of collecting markers of neural function like EEG power and SRT in low-resource settings. Including such measures in future trials in LMICs is important for improving understanding of the impact of early interventions on the brain and illuminating potential neural mechanisms through which interventions affect behavior and skills [2,7].

## Supporting information

**S1 CONSORT Checklist. CONSORT checklist.**
(DOCX)

**S1 Text. Trial protocol.**
(DOCX)

**S2 Text. Additional detail on lab assessment procedures.**
(DOCX)

**S1 Table. Template for intervention description and replication (TIDieR) checklist.**
(DOCX)

**S2 Table. CONSERVE-CONSORT checklist.**
(DOCX)

**S3 Table. Characteristics of EEG and SRT observations.**
(DOCX)

**S4 Table. Impact on MDAT z-scores in lab subsample.**
(DOCX)

**S1 Fig. Intervention job aids.**
(PDF)

## Acknowledgments

The authors would like to thank local collaborators at the Anova Health Institute, the Limpopo Province Department of Health, and the Mopani District Department of Health who provided support in the design and piloting of the intervention and in implementation of the trial.

## Author Contributions

**Conceptualization:** Peter C. Rockers, Jukka M. Leppänen, Amanda Tarullo, Günther Fink, Davidson H. Hamer, Aisha K. Yousafzai, Denise Evans.

**Data curation:** Jukka M. Leppänen, Amanda Tarullo, Lezanie Coetzee.

**Formal analysis:** Peter C. Rockers, Günther Fink.

**Funding acquisition:** Peter C. Rockers, Jukka M. Leppänen, Amanda Tarullo, Denise Evans.

**Methodology:** Peter C. Rockers, Jukka M. Leppänen, Amanda Tarullo, Günther Fink, Davidson H. Hamer, Aisha K. Yousafzai, Denise Evans.

**Project administration:** Peter C. Rockers, Jukka M. Leppänen, Amanda Tarullo, Lezanie Coetzee, Denise Evans.

**Supervision:** Jukka M. Leppänen, Amanda Tarullo, Lezanie Coetzee, Denise Evans.

**Validation:** Jukka M. Leppänen, Amanda Tarullo.

**Writing – original draft:** Peter C. Rockers.

**Writing – review & editing:** Peter C. Rockers, Jukka M. Leppänen, Amanda Tarullo, Lezanie Coetzee, Günther Fink, Davidson H. Hamer, Aisha K. Yousafzai, Denise Evans.

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
