## [Editor Report · Decision Letter 0]

3 Oct 2022

Dear Dr Rockers, 

Thank you for submitting your manuscript entitled "Impact of a community health worker home visit intervention on child development in South Africa: A cluster-randomized controlled trial" for consideration by PLOS Medicine.

Your manuscript has now been evaluated by the PLOS Medicine editorial staff and I am writing to let you know that we would like to send your submission out for external peer review.

Please re-submit your manuscript within two working days, i.e. by Oct 05 2022 11:59PM.

Kind regards,

Beryne Odeny

PLOS Medicine

---

## [Decision Letter · Decision Letter 1]

22 Nov 2022

Dear Dr. Rockers,

Thank you very much for submitting your manuscript "Impact of a community health worker home visit intervention on child development in South Africa: A cluster-randomized controlled trial" (PMEDICINE-D-22-03227R1) for consideration at PLOS Medicine. 

Your paper was evaluated by an associate editor and discussed among all the editors here. It was also discussed with an academic editor with relevant expertise, and sent to independent reviewers, including a statistical reviewer. The reviews are appended at the bottom of this email and any accompanying reviewer attachments can be seen via the link below:

[LINK]

In light of these reviews, I am afraid that we will not be able to accept the manuscript for publication in the journal in its current form, but we would like to consider a revised version that addresses the reviewers' and editors' comments. Obviously we cannot make any decision about publication until we have seen the revised manuscript and your response, and we plan to seek re-review by one or more of the reviewers. 

We hope to receive your revised manuscript by Dec 13 2022 11:59PM. Please email us (plosmedicine@plos.org) if you have any questions or concerns.

We look forward to receiving your revised manuscript. 

Sincerely,

Callam Davidson, 

PLOS Medicine

plosmedicine.org

Comments from the Academic Editor:

The authors need to address all the 4 reviewers’ comments in detail.

They should specifically address the issue of training CHWS and their supervisors. Additionally, the authors should:

1) give more information on program implementation, especially in the Methods section.

2) revisit and give details of sample size determination including the power. How were missing data handled?

3) revise the conclusions section and make sure that they are based on the study findings.

4) clearly show the readers how to access the trial registration details on- PACTR201710002683810 and include the relevant link.

5) give details on reliability of the MDAT.

Abstract Methods and Findings:

* Please ensure that all numbers presented in the abstract are present and identical to numbers presented in the main manuscript text.

* Please specify who was blinded to the intervention and control.

* Please define the intervention and control states.

* Please include the important covariates that are adjusted for in the analyses.

* Please provide the number of participants lost to follow up in each group.

Please conclude the Introduction with a clear description of the study question or hypothesis.

Please remove the ‘Role of the funding source’ section.

Please cite the items in your Supporting Information at the relevant points in the main text (guidance for how to do this can be found here: https://journals.plos.org/plosmedicine/s/supporting-information)

Please specify whether informed consent was written or oral.

Please define "lost to follow-up" as used in this study. Other reasons for exclusion should be defined.

Thank you for including the CONSORT checklist as Supporting Information. Please update the checklist to use section and paragraph numbers, rather than page numbers.

The key exclusion criteria in the Methods differ from those described in the PACTR registry. Please explain, including in the manuscript.

The secondary outcomes measures are not described in the study protocol. If the outcomes were not prespecified in the protocol, please indicate that they were post hoc and explain why they were added. Post hoc comparisons should be presented as hypothesis generating rather than conclusive.

The trial registration lists the outcomes of ‘autonomic assessments of each child’s heartbeat and breathing’ and ‘blood samples from children tested for biomarkers of systemic inflammation’. The trial registration also states that ‘an additional sample of 50 additional caregiver/child pairs of high socioeconomic status (SES) will be recruited from Greater Tzaneen and assessed with EEG and ET technologies to explore the relationship between household wealth and child neurodevelopment’. (a) Can you please present those results as part of this manuscript, or indicate why that is not possible? (b) Can you please indicate when you plan to publish those results?

Please define the abbreviation in Figure 3 (saccadic reaction time).

Lines 459-460: As your trial had to undergo important modifications in response to extenuating circumstances, please complete the CONSERVE-CONSORT checklist and provide in your Supporting Information.

Did your study have a prospective analysis plan? Please state this (either way) early in the Methods section.

a) If a prospective analysis plan was used in designing the study, please include the relevant prospectively written document with your revised manuscript as a Supporting Information file to be published alongside your study, and cite it in the Methods section. A legend for this file should be included at the end of your manuscript. 

For reference 21, please include the date cited. 

RE reference 44 listed as in preparation, papers cannot be listed in the reference list until they have been accepted for publication or are publicly available on a preprint archive. The information may be cited in the text as a personal communication with the author if the author provides written permission to be named. Alternatively, please provide a different appropriate reference. 

Comments from the reviewers:

Reviewer #1: Statistical review

This paper reports a cluster randomised trial assessing the effectiveness of a community health worker delivered intervention on child development.

Generally the trial used suitable statistical methods and was reported well, with changes from protocol were clearly explained. I had some minor comments which are provided below:

1. Abstract "had no impact" could be "had no significant impact" (and similarly for subsequent results). I would recommend using alternative phrasing to 'a beta' - perhaps mean difference? 

2. Abstract/outcomes: It seemed that some secondary outcomes (e.g. child diet) were not reported in the abstract. Generally it is best practice to report all secondaries (or at least add 'no significant effect for other secondary outcomes) or none.

3. Trial registration: I could not find PACTR201710002683810 on the PACTR website - can a link be provided? I would recommend all secondary outcomes mentioned in the registration page are reported in the paper (or a reason given why they cannot be).

4. Statistical analysis: I would recommend the text about the variables that were adjusted for is moved to earlier in the statistical analysis section (when read the early parts I had thought it had been missed out).

5. Statistical analysis: The primary analysis would account for missing at random data. Since the proportion of participants who dropped out was relatively high it would be good to mention how missing data was dealt with. The discussion might mention whether the missing at random assumption is plausible.

James Wason

Reviewer #2: Thank you very much for requesting a review of this article. 

This paper presents a cluster randomised trial of a home visiting intervention integrated into community health worker practice in South Africa looking at an intervention that provided health workers with a job aid that included information on child health, nutrition, developmental milestones and encouragement to engage in developmentally appropriate play based activities on monthly visits for children less than 2 years of age. This is linked to the government's proposed National Integrated Early Childhood Development Policy programme. 

The team measured household data as well as anthropometry, child development using MDAT, EEG and saccadic reaction time. It is an interesting study and demonstrates a few important results that should be published - in particular the early changes in neural testing on EEG and saccadic reaction time. 

Generally, it is a well written and clear paper and I have few queries - although would recommend that a statistician review the cluster trial analysis. My points are outlined below and mainly relate to representativeness of the sample tested:

1. Recruitment occurred through "active monitoring and review of health facility records" - do we have an idea as to what "reach" this created and whether this identified a representative sample of the population? Table 1 provides info on study participants including those enrolled and those who had endline assessments - is it clear if there were any significant differences between these groups? 

2. Is there any information about whether those who came for neural testing were representative of the sample as well as the population? Line 351 makes out that there were no clear differences in baseline demographics comparing observations included to those excluded - does this mean those included for neural testing vs excluded from neural testing had similar demographics. I think this might be the case but the way it was written was not entirely clear. 

3. Lines 127 - 131 - Are there any reasons why there was a change in emphasis to "responsiveness" in the activities promoted in the guides? Could this be explained better?

4. MDAT assessments - any information on the reliability of these assessments and any reliability done prior to starting these assessments? Was it translated and back translated into local language? 

5. The abstract makes out that the significant improvements in EEG and saccades (neural testing) were throughout all time points (or at least it does not specify). I wonder if it might be important in the abstract to try and be clearer that the significant findings were only there in wave 1 and 2 and not in wave 3 and that similarly anthropometry and MDAT were only measured in wave 3. 

Reviewer #3: 

Thank you for this interesting paper, which looks at the impact of a CHW home visiting intervention on several outcomes, including linear growth, development and neural function through electroencephalography (EEG) and saccadic reaction time (SRT) testing, using a cluster randomized control trial design. 

I found the approach of using the assessment of children's neural function as an outcome measure interesting, unusual and innovative, but feel that the detailed focus in the methods and outcomes sections of the paper on the EEG and SRT assessment has resulted in the omission of critical information for your readers about the intervention itself and its theoretical basis. There is also no information about what type of visits the mother-child dyads in the control group received - my understanding from the South African WBOT system is that CHW are expected to perform several postnatal home visits (in the methods section you mention that all children under 2 years are expected to be visited at home monthly). In the light of this, could you please answer the following: 

1) How does the intervention compare to the standard care received in the control group? Describe the usual standard care in the control group and whether this was monitored in any way. There needs to a paragraph or a subsection in the methods clearly contrasting the intervention group with the control group and showing your reader the difference between the two groups. 

2) I assume that randomization of WBOT clusters of CHW was performed before training with the "job aid" occurred and that only CHW that were on the intervention side received the training? If this is the case, it would be good to mention this under the methods section. It is not clear from the flow chart (Fig2) of the "Trial profile" how many CHW were in the control group and how many were in the intervention group. Please mention this in the text and add this into the figure. 

3) Further - in figure 2 - for the lost to follow up boxes it would be better to use the term "dyads" rather than "participants" to remain internally consistent. Under deaths, it please differentiate between maternal and child/infant deaths. 

4) Were all the CHW proficient in English? Did trainings occur in English or in Sepedi or Xitsonga? How long was the training period for the tool? You mention that didactic trainings occurred, and that live demonstrations of play activities were shown, but was there any in-the-field training and supervision of CWH who were using the job aid? For those CHW who were not proficient in English, were there translated versions of the job-aid available or was this not deemed necessary? 

5) The job aid looks like an amazing resource which would be useful to CHW all over South Africa. Is it available for use by and distribution to government and NGO CHW in other provinces? Would it be possible to make it available for the patient-facing page to be translated into other South African languages? 

6) As a rural clinician, I can see the practical utility of the job aid for CHW as a straightforward, simple and well-structured tool to guide CHW visits in homes. It would be good for you to motivate in the paper why you regard it as a useful tool that might make a difference to child outcomes. Was there any theoretical/evidence basis for its contents and structure? 

7) It isn't clear from the paper whether the main focus of the study was to test whether the job aid was a useful intervention tool that would help CHW and improve child outcomes, or whether your focus was on seeing whether the neurological assessments were practically possible to do and helpful from an outcome perspective. This needs to be clarified in the text. 

8) It is a pity that only about half the clusters were assessed using EEG and SRT. Was this because of cost or because of distance or time? It would be good to explain why only half to a third of the children recruited into the trial were assessed in this way

9) How long did each full neurological assessment take? What is the level of expertise required by operators of the EEG and SRT equipment? Does this need to be done by a medical doctor or can it be done by a lay worker who has been trained? Again, I think this would be useful information for your readers. 

10) I assume that you attempted to recruit all children born in the catchment area of each WBOT after 15 December 2017 to ensure that there was no selection bias - could you expand a bit more about exactly how mother-child dyads were recruited into the study? You mention "active monitoring by CHWs" and that health facility records were reviewed - were these hospital and clinic birth registers or other clinic or hospital records? Please include this in the manuscript. 

11) You mention that once details of children born were found, dyads were recruited in their homes and that household survey data were also collected at baseline. What was the average age of the infants at baseline? And was any anthropometric data collected at this baseline visit? 

12) Under the "Data collection" (line 142) of the Methods section there is no mention of when anthropometric measurements were performed, and whether these were once-off or serial measurements. The data collection for secondary outcomes, including caregiver engagement, dietary diversity score and household wealth index and should also be included here and not later, under "Outcomes", (line 295 to 298) as it is currently. 

13) At line 345 and 346 you mention that 266 intervention group dyads and 224 control group dyads were recruited for the lab portion of the study. Please put in the TOTAL number recruited to the lab section (490) in here too. There was a big drop off from recruitment into the lab portion at baseline to the first lab visit around 7 months (from 490 to 316) - but no drop out from the first to the second lab visit and only a small drop off to the endline assessment. Do you have any understanding why there was such a big, initial loss to follow up? Were participants nervous of the EEG testing etc. and therefore decide to withdraw or were there any other clear causes of this drop off?

14) As I have mentioned earlier, the neurological assessments you performed in the lab are an interesting and innovative, but as a public health clinician and researcher, I found the explanations very wordy, complicated and difficult to understand. It would be good to, in the introduction, write a little bit more about the utility and potential benefits of neural testing for early life interventions in simple, accessible language. And in the methods and results sections, it would also be useful to have a short, clear summary at the end of each technical section that cuts through the jargon and makes the EEG/SRT findings understandable by non-neurologists and non-specialists. 

15) In the discussion, you mention that SRT was significantly reduced, indicating positive impact. However, while the EEG the impact on gamma power was significant, it was in the opposite direction to what you expected, and you reference an article indicating that this measure might not be the most appropriate one to use in this context because of the impacts of nutritional deprivation on scull conductivity. Why not just recommend SRT on its own? Your findings seem to indicate that EEG findings are confusing and counter to expectations. In the absence of good literature that suggests alternative, appropriate EEG measures in the rural African context, would a fairer conclusion of your study not be that SRT seemed helpful, yet EEG findings were counter to expectation and cannot, on the basis of this study, be recommended for use in the low-income African context util more appropriate measures are found? 

16) It appears that only one height for age measure and Z score was calculated on each child at endline? Is there any reason why you decided not to perform serial height measurements over time - which may have given a better picture of the growth of the children investigated? 

17) As one of your conclusions, you indicate that you believe that preforming EEG and SRT measurements are feasible in the rural African context. It would be useful if readers could get a sense of the time, expertise required and what the cost of running these neural tests would be. 

18) Finally - SRT in particular appears to be a more sensitive marker of neurological and developmental impact than other traditional measurements such as growth or developmental markers, but it is a pity that these positive SRT effects seem to have faded a year after the intervention ended. Do you think that CHW should continue household interventions with beyond 2 years - and if so, how regularly and for how long? This may be worth raising in your discussion and conclusions. 

Thanks again for this interesting paper. I am hopeful that your work will help researchers to investigate the impact of home visiting CHW interventions with more sensitive tools. 

Reviewer #4: Review of the Manuscript: PMEDICINE-D-22-03227R1 

Impact of a community health worker home visit intervention on child development in South Africa: A cluster-randomized controlled trial

This manuscript addresses a very important topic, which is the effectiveness of an early childhood home visiting intervention implemented within an existing community health care system. While such programs have been found to improve developmental outcomes in research studies, further work is needed to understand how they can be scaled up with high fidelity and maintain effectiveness. Given the importance of this question, it is unfortunate that there is very little information on program implementation in the manuscript. Information on measures of program coverage are reported in unexpected places, such as the statistical analysis section and the Discussion, instead of the Methods where they would be expected to be reported. Other information on key factors such as implementation fidelity, workforce capacity, workforce coverage, and equity of access to the program are missing. Further comments on implementation information:

- Page 10, line 135 "The job aid was designed to be effort neutral by restructuring the already allotted time." This sentence is not clear. Does it mean that adding the use of the job aid during the visit did not change the amount of time the CHW was intended to spend at the household? How much time was allotted? What were the other activities the CHW were meant to do at the visit before adding the job aid? How did the structure of the visit change with the addition of the job aid?

- Page 10, lines 135-137 Who were the trainers? Were the trainers members of the research team or was the training also integrated into the existing system? If the trainers were research personnel, how would the program be implemented on a large scale within the existing system? What about supervisors? How were the supervisors trained and how did they monitor CHW performance? What about their level of education and CHW level of education? All of this is very important to consider for scaling up the intervention. 

- Page 18-19, lines 320-323. The description of how CHW compliance was measured belongs earlier in the description of data collection methods, not in the Statistical Analysis section.

- Were any other data collected on fidelity of program implementation? C.A.R.E. (consolidated advice for reporting ECD implementation research) guidelines recommend: "Describe the methods and tools used to assess fidelity of intervention delivery and receipt of intervention by recipients (e.g., self-report, observation, competency tests, checklists, and monitoring records on recipient participation), sources of data, and at which stage of the project implementation was fidelity assessed. If relevant, describe whether these data were collected in the comparison conditions." Other aspects of the CARE guidelines would also be beneficial to improve the manuscript.

Yousafzai, A.K., Aboud, F.E., Nores, M. and Kaur, R. (2018), Reporting guidelines for implementation research on nurturing care interventions designed to promote early childhood development. Ann. N.Y. Acad. Sci., 1419: 26-37. https://doi.org/10.1111/nyas.13648

Several issues regarding the sample selection and sample size need to be addressed:

- It is not clear how many CHWs were employed in the catchment area, how many of those were invited to participate, and of those invited, how many did participate. Therefore, it is difficult to evaluate whether the sample enrolled in the trial was likely to be representative of the population. 

- Were the authors able to conduct any assessment of whether the sample enrolled in the trial was likely to be representative of the population?

- Page 18, lines 301-303. For a cluster-randomized trial, the power analysis must be based on the number of clusters, not the total number of individuals. The sample-size determination usually involves either the determination of the number of clusters given cluster size or the determination of cluster size given the number of clusters. Either way, it must be stated the number of clusters on which the power analysis was based.

- What was the power for the sub-sample?

Abstract:

- First sentence of abstract "achieving scale" is not clear what this means.

- Methods: Please state the target n of the EEG subsample. Clarify whether the 'lab subsample' is the same as the EEG subsample. Was EEG the only assessment conducted at the lab visit? Where were other outcome assessments conducted? It's not clear which outcomes were assessed on the full sample and which were assessed on the subsample.

- Findings: Please state the attrition of clusters as well as individuals (at endline, were there still 51 clusters)?

- Conclusions: The statement "we found evidence of an effect on neural function" is confusing and misleading because the effect on SRT indicated a positive effect of the intervention on neural maturity, while the effect on EEG power indicated a negative effect of the intervention on neural maturity, which is not clear in the abstract.

- The statement "Strengthening supervision of CHWs to improve compliance is necessary to maximizing impact" does not follow from any of the results presented in the Abstract.

Other comments: 

Page 4, line 10 Evidence on their effectiveness to do what?

Page 4, line 16 "requires careful planning" is not clear. Does this mean the program directors/supervisors of the CHWs need to plan carefully how to implement the program? Or does this mean the CHWs need to plan carefully to manage their daily schedules? Or something else?

Page 5, line 41-42: How many CHWs were invited to participate?

Page 6, line 61: How many? What was the target sample size for the subsample? Were all mother-child dyads in the target sub-district invited?

Page 20, line 351-352. In this sentence, it is not clear what "observations included" and "those excluded" is referring to. Is this to assess potential bias in loss to follow up for the main outcomes or potential bias in the subsample? I don't see any information on the former in supporting information. It is also difficult to assess this from Table 1 because there is no information on the group lost to follow-up specifically.

Table S2. It is not clear where the number included and excluded is coming from. These numbers in Table S2 do not match the numbers in Figure 2 Trial Profile.

Page 23, line 409-411. In the per protocol analysis, what proportion of the control group was used as the comparison group and how was this selected?

Page 28, line 504-505. The conclusion that strengthening supervision of CHWs is necessary to maximizing impact does not follow from the evidence of the study because no information on supervision was presented or analyzed.

Page 28, line 505-506. It is not clear what evidence is the basis of the conclusion that this study demonstrates the feasibility of collecting neural markers like EEG and SRT. Is it based on the result that 91% of EEG observations and 78% of SRT observations were met the pre-set validity criteria for inclusion in the analysis? This result was not discussed or interpreted. Is this what would be expected to demonstrate the feasibility of the method? Or is this conclusion based on a different result?

[LINK]

---

## [Decision Letter · Decision Letter 2]

8 Mar 2023

Dear Dr. Rockers,

Thank you very much for re-submitting your manuscript "Impact of a community health worker home visit intervention on child development in South Africa: A cluster-randomized controlled trial" (PMEDICINE-D-22-03227R2) for review by PLOS Medicine.

I have discussed the paper with my colleagues and the academic editor and it was also seen again by two reviewers. I am pleased to say that provided the remaining editorial and production issues are dealt with we are planning to accept the paper for publication in the journal.

[LINK]

We look forward to receiving the revised manuscript by Mar 15 2023 11:59PM.   

Sincerely,

Callam Davidson, 

Senior Editor 

PLOS Medicine

plosmedicine.org

Requests from Editors:

Please add line numbering to your manuscript. 

Please update your title to ‘Evaluation of a community health worker home visit intervention on child development in South Africa: A cluster-randomized controlled trial’

Please consider including additional intervention resources as Supporting Information or uploading them to an online repository and citing this in the manuscript. 

Please provide Figure 1 as a separate file per our guidelines: https://journals.plos.org/plosmedicine/s/figures

Suggestions for shortening the Abstract:

* Delete ‘’but masking of participating CHWs and dyads was not possible’.

* Shorten the description of the intervention and control states (the latter can be described as standard of care).

* The list of covariates can be removed and described simply as ‘a set of demographic covariates’.

* Delete ‘175 dyads not assessed at endline’.

* Details of data collection can be trimmed (current abstract text is duplicated in the Methods).

* The Abstract Conclusions can be shortened. Please interpret the study based on the results presented in the abstract, emphasizing what is new without overstating your conclusions.

Please avoid recycling text from the Abstract in your Author Summary, which is intended for a non-specialist audience. 

Author Summary: Please include headline numbers, including sample size and key findings. 

Your Author Summary mentions '764 study children' but this is not mentioned elsewhere in the manuscript. Please ensure numbers are consistent throughout the text. 

Please consider moving some of the more technical aspects of the Methods to the Supporting Information (for instance, lab assessments).

Please replace the term “compliance” with “adherence”.

Please ensure all Internet sources include a date of citation (e.g., #30). 

Comments from Reviewers:

Reviewer #1: Thank you to the authors for addressing my previous comments well. Just two minor clarifications/recommendations: 1) if the analysis used a complete cases approach then the assumption would be missing completely at random (MCAR) rather than missing at random (MAR); 2) I would recommend that non pre-specified outcomes are referred to as exploratory outcomes rather than secondary.

Reviewer #4: The revised manuscript is much improved. A few additional comments:

Line 168-169. "The job aid used in the intervention arm expanded the content of home visits but was designed to be effort neutral by restructuring the time CHWs were already responsible for allotting." The difference between the intervention and control groups is still not clear. What activities or information were delivered in the control group that were not delivered in the intervention group? A table comparing the content and activities covered by the CHWs in each group would be useful.

Line 528-592. "To our knowledge, no previous trial in an LMIC has examined the impact of an early-life intervention on EEG power or SRT." Please see:

Larson, L. M., D. Feuerriegel, M. I. Hasan, S. Braat, J. Jin, S. M. M. U. Tipu, S. Shiraji, F. Tofail, B.-A. Biggs, J. Hamadani, K. Johnson, S.-R. Pasricha and S. Bode (2023). "Supplementation With Iron Syrup or Iron-Containing Multiple Micronutrient Powders Alters Resting Brain Activity in Bangladeshi Children." The Journal of Nutrition 153(1): 352-363.

Prado, E. L., K. Maleta, B. L. Caswell, M. George, L. M. Oakes, M. C. DeBolt, M. G. Bragg, C. D. Arnold, L. L. Iannotti, C. K. Lutter and C. P. Stewart (2020). "Early Child Development Outcomes of a Randomized Trial Providing 1 Egg Per Day to Children Age 6 to 15 Months in Malawi." J Nutr 150(7): 1933-1942.

Response time on the IOWA task is a type of saccadic reaction time.

[LINK]

---

## [Editor Report · Decision Letter 3]

21 Mar 2023

Dear Dr Rockers, 

On behalf of my colleagues and the Academic Editor, Professor James Tumwine, I am pleased to inform you that we have agreed to publish your manuscript "Evaluation of a community health worker home visit intervention to improve child development in South Africa: A cluster-randomized controlled trial" (PMEDICINE-D-22-03227R3) in PLOS Medicine.

When making the formatting changes please address the following editorial requests:

1) Author Summary:

* Bullets 1 and 2 under question 2 can be combined. 

* Please define the control group as well as the intervention group.

* Please simplify the bullet point in which you define the sample size.

2) Figure 3: Please include a break in the y-axis to indicate the scale does not begin at zero. 

3) Please ensure the updated version of S2 text (with figures redacted) is included with the final submission.

PRESS

Sincerely, 

Callam Davidson 

Associate Editor 

PLOS Medicine